# FREE-STYLE AND FAST 3D PORTRAIT SYNTHESIS

## ABSTRACT

Efficiently generating a free-style 3D portrait with high quality and consistency is a promising yet challenging task. The portrait styles generated by most existing methods are usually restricted by their 3D generators, which are learned in specific facial datasets, such as FFHQ. To get a free-style 3D portrait, one can build a large-scale multi-style database to retrain the 3D generator, or use a off-the-shelf tool to do the style translation. However, the former is time-consuming due to data collection and training process, the latter may destroy the multi-view consistency. To tackle this problem, we propose a fast 3D portrait synthesis framework in this paper, which enable one to use text prompts to specify styles. Specifically, for a given portrait style, we first leverage two generative priors, a 3D-aware GAN generator (EG3D) and a text-guided image editor (Ip2p), to quickly construct a few-shot training set, where the inference process of Ip2p is optimized to make editing more stable. Then we replace original triplane generator of EG3D with a Image-to-Triplane (I2T) module for two purposes: 1) getting rid of the style constraints of pre-trained EG3D by fine-tuning I2T on the few-shot dataset; 2) improving training efficiency by fixing all parts of EG3D except I2T. Experimental results show that our method is capable of synthesizing high-quality 3D portraits with specified styles in a few minutes, outperforming the state-of-the-art.

## 1 INTRODUCTION

Portrait synthesis (Karras et al., 2019; 2020; Gu et al., 2022; Chan et al., 2022) is a promising yet challenging research topic for its wide range of application potential, e.g. game character production, Metaverse avatars and digital human. With the rapid development of generative models such as generative adversarial models (Goodfellow et al., 2014), 2D portrait synthesis has achieved remarkable success. After that, many methods (Karras et al., 2019; 2020; 2021) are proposed to improve the generation quality to photo-realistic level.

Recently, 3D portrait synthesis has attracted more and more attention, especially with the emergence of Neural Radiance Field (NeRF) (Mildenhall et al., 2020). As the representatives among them, 3D-aware GAN methods (Gu et al., 2022; Chan et al., 2022; Or-El et al., 2022) combine NeRF with StyleGANs (Karras et al., 2020) to ensure 3D consistency synthesis. By mapping an image to the 3D GAN latent space, 3D GAN inversion approaches (Ko et al., 2023; Lin et al., 2022; Yin et al., 2022) can generate or edit a specific 3D portrait. However, both of them fail to create a free-style 3D portrait, e.g., a style defined by user's text prompt, since their generators are usually trained on a dataset that follows a particular style distribution, such as the realism style in FFHQ (Karras et al., 2019), which raises a question: how to generate a free-style 3D portrait at a low cost? One may collect a large number of portrait images with different styles to retrain their models, but the data preparing and training process are usually time-consuming. Another potential solution is that synthesizing a style-specific 3D portrait first, then transferring it to any style with a off-the-shelf style transfer tool. Unfortunately, the 3D consistency will be difficult to be maintained.

To this end, we propose an efficient pipeline to achieve free-style 3D portrait synthesis in this paper. At first, we leverage the knowledge of two pre-trained generative priors, EG3D (Chan et al., 2022) and Instruct-pix2pix (Ip2p) (Brooks et al., 2022) to construct a few-shot portrait dataset with a given style, avoiding dirty data collection and cleaning. The former generates a multi-view 3D portrait, and the latter performs text-guided style editing in each viewpoint. We empirically find the editing results of Ip2p vary significantly along viewpoints for some given style, resulting in the issue of multi-view misalignment. To alleviate this problem, an optimization strategy is introduced into

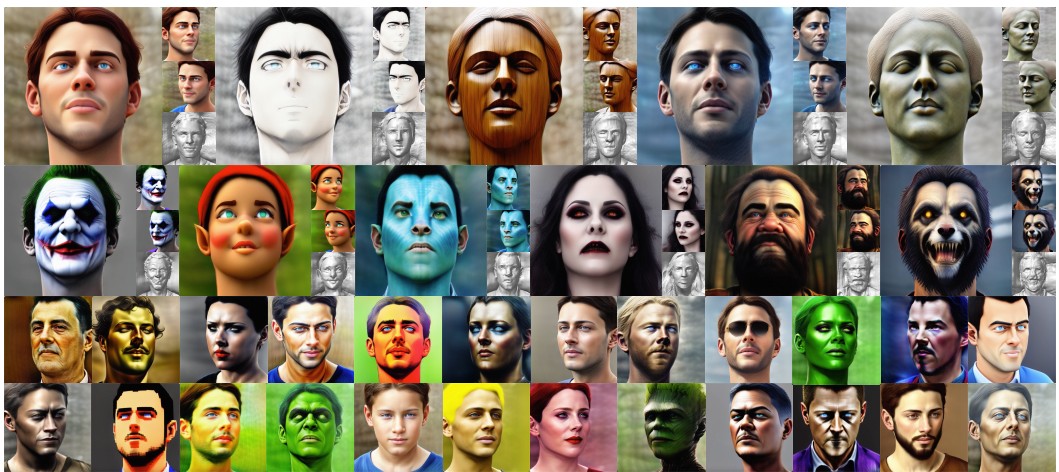

Figure 1: Our 3D portrait synthesis results. The first row shows man with different styles, the second row shows various characters, and the last two rows show diverse identities with different styles from varied viewpoints.

the inference stage of Ip2p. Secondly, we replace the original triplane generator with a trainable I2T network, and freeze other parts of EG3D to avoid training the total EG3D. We pre-train the I2T network to create the mapping from portrait image to triplane feature space, which helps I2T network fine-tune the few-shot set in a few minutes. Our high-quality stylized 3D portrait synthesis results are shown in Fig. 1.

## 2 RELATED WORK

### 2.1 3D-AWARE GAN

With the development of neural implicit representation (NIR) represented by neural radiance fields (NeRF) (Mildenhall et al., 2020), more and more methods (Michalkiewicz et al., 2019; Niemeyer et al., 2019; Chibane et al., 2020b; Atzmon & Lipman, 2020; Chabra et al., 2020; Jiang et al., 2020; Chibane et al., 2020a; Gropp et al., 2020) are focusing on learning 3D scenes and 3D object representation using neural networks. NeRF represents the 3D scene as a series of neural radiance and density fields, and uses volume rendering Kajiya & Von Herzen (1984) technique for 3D reconstruction. Similarly, some methods Sitzmann et al. (2019); Niemeyer et al. (2020) learn neural implicit representation using multi-view 2D images without 3D data supervision. However, even multi-view data is usually expensive to construct in some scenes, such as portraits, so many approaches gradually migrate to learn 3D-aware GAN using unstructured data, i.e., single-view portraits, based on the idea of adversarial training. PiGAN Chan et al. (2021) proposes a siren-based neural radiance field and uses global latent code to control the generation of shapes and textures. GIRAFFE Niemeyer & Geiger (2021) proposes a two-stage rendering process, which first generates the low-resolution features with a volume renderer, and then learns to upsample the features with a 2D CNN network. Some methods introduce StyleGAN structures into the 3D-aware GAN. StyleNeRF Gu et al. (2022) integrates NeRF into a style-based generator to improve rendering efficiency and 3D-consistency of high-resolution image generation. StyleSDF Or-El et al. (2022) merges a Signed Distance Fields representation with a style-based 2D generator. EG3D Chan et al. (2022) proposes a triplane 3D representation method to improve rendering computational efficiency and generation quality. Some approaches have also started to focus on the control and editing of 3D-aware GANs. FENeRF (Sun et al., 2022) and Sem2nerf(Chen et al., 2022) introduce semantic segmentation into the generative network, and learn a whole neural radiance field with semantic information. CNeRF (Ma et al., 2023) proposes a compositional neural radiance field to split the portrait into multiple semantic regions, and learns semantic synthesis separately with a local neural radiance field, and finally fuses them into a complete 3D representation of the portrait. Along with the development of 3D-aware GAN, 3D GAN Inversion methods (Ko et al., 2023; Lin et al., 2022; Yin et al., 2022) have appeared.

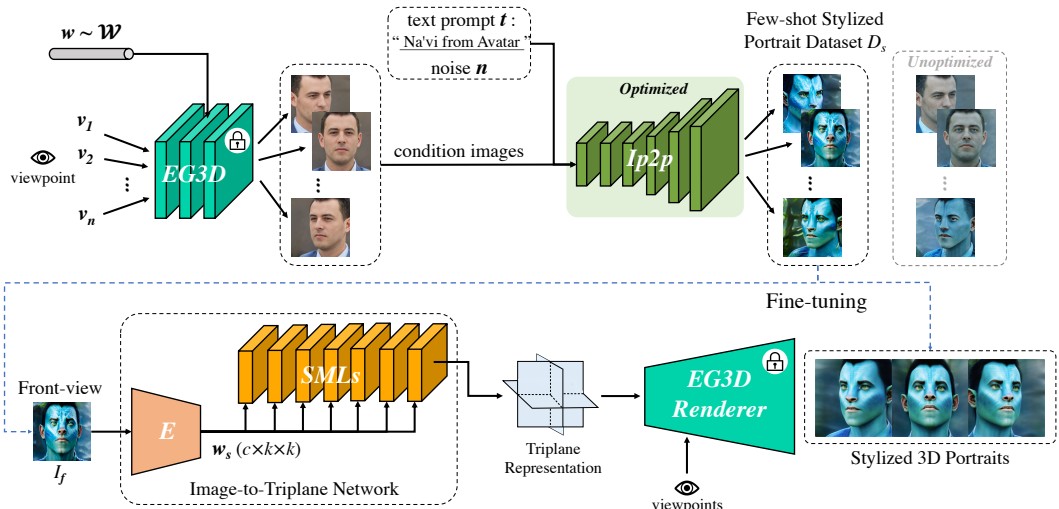

Figure 2: The framework of our method. We introduce two generative priors EG3D and Ip2p to quickly build a few-shot stylized portrait dataset $\mathcal{D}_s$. Ip2p is optimized to produce more stable and consistent text-guided stylization results. With the constructed few-shot dataset, we quickly fine-tune a pre-trained I2T network to achieve stylized 3D portrait synthesis.

They learn to map real images to the latent space of 3DGAN for image inversion and editing. However, such methods face a problem that they cannot jump out of the pre-trained 3DGAN prior and cannot synthesize out-of-distribution portraits. In this paper, we propose a new framework that can synthesize stylized 3D portraits freely, which is not restricted by the 3D generative prior and can generate 3D portraits of specific styles based on text prompts.

## 2.2 TEXT-GUIDED IMAGE EDITING

There are numerous image editing methods, and the performance of text-guided image editing methods (Avrahami et al., 2022; Hertz et al., 2022; Kawar et al., 2023; Brooks et al., 2022; Haque et al., 2023; Liu et al., 2023) has been qualitatively improved thanks to the advancement of pre-trained image generation large models (Rombach et al., 2022; Ramesh et al., 2021; 2022) based on the Diffusion model. Ip2p (Brooks et al., 2022) is a SOTA text-guided image editing method, which uses two generative priors, GPT-3 Brown et al. (2020) and Stable Diffusion (Rombach et al., 2022), to synthesize a large number of paired images and then train a conditional diffusion model on them. This model allows the users to provide a relatively free text instruction to edit a given image, including stylistic transfer. Therefore, Ip2p is well suitable as a text-guided image editing prior for this paper to perform text-guided style transfer on portraits from different viewpoints. However, the model also has problems, such as poor generation with some simple text prompts and generating portraits with large stylistic variations for different views of the same portrait. We propose some improvements to solve these problems in this paper.

## 3 METHODOLOGY

In this section, we detail our free-style and fast 3D portrait synthesis framework, as shown in Fig. 2. We first briefly introduce two generative priors, EG3D and Ip2p, and combine them to build a few-shot dataset with a given style. To describe styles more freely, we optimize Ip2p to make it more stable for stylizing portraits from different perspectives (Sec. 3.1). We then use the few-shot dataset to fine-tune our proposed I2T network, which is equipped with EG3D prior to achieve fast stylized 3D portrait synthesis (Sec. 3.2).

## 3.1 FEW-SHOT DATASET CONSTRUCTION

We denote the input style as $\mathbf{t}$, which is often defined by text prompt from user.

**3D-aware GAN prior.** As a state-of-the-art 3D-aware GAN method, EG3D (Chan et al., 2022) can be expressed as $G(\theta, \mathbf{w}, \mathbf{v})$, where $\theta$ is the model parameters, $\mathbf{w}$ is a sampling vector in the $\mathcal{W}$ latent space, and $\mathbf{v}$ is the view direction to be rendered. We randomly sample a $\mathbf{w}$ vector, and set $\mathbf{v}$ as follows: assuming pitch and yaw angles of front view portrait are zero, $\mathbf{v}$ is uniformly sampled $i$ times within both pitch and yaw range of $-30°$ degrees to $30°$ degrees. We denote the candidate set of $\mathbf{v}$ as $(P, Y)$, which contains $i^2$ sampling results ($i\ pitch \times i\ yaw$). Then $G$ can output $i^2$ portrait images along each $\mathbf{v}$. Note that these portraits keep the same identity since $\mathbf{w}$ is only sampled once.

**Text-guided image editing prior.** Next, we employ Ip2p (Brooks et al., 2022) to perform the style editing on the portraits generated above. Ip2p implements conditional image editing based on Stable Diffusion (Rombach et al., 2022), which can be denoted as $T(\phi, I, \mathbf{n}, \mathbf{c})$, where $\phi$ is the model parameters, $I$ is the input portrait, $\mathbf{n}$ is the Gaussian noise used in the denoising process, and $\mathbf{c}$ is the text prompt that guides the editing direction. Considering different input noise will generate different results, we fix $\mathbf{n}$ to keep the identity unchanged. Then we set $\mathbf{c} = \mathbf{t}$, let $I$ be the portraits that are produced by EG3D, and use $T$ to generate stylized portraits.

**Optimizing Instruct-pix2pix inference.** As mentioned above, some $\mathbf{t}$ will cause Ip2p to generate unsatisfactory stylized portraits. For example, the style "Na'vi from Avatar" will make the portrait style vary greatly from one viewpoint to another. More samples are listed in Sec. 4.4. Therefore, we want to optimize Ip2p to make it generate stable results for different $\mathbf{t}$. Inspired by SDEdit (Meng et al., 2021), we replace the original Gaussian noise $\mathbf{n}$ with a new noise $\mathbf{n}^*$ during the inference stage of Ip2p:

$$\mathbf{n}^* = Add(\mathcal{E}(I), \mathbf{n}, \tau), \tag{1}$$

where $\mathcal{E}(I)$ denotes the latent features obtained from the Stable Diffusion encoder of $I$, $\tau$ is the degree of noise addition, and $Add$ represents the standard DDPM (Ho et al., 2020) noise addition operation. In addition, we design an enhanced prompt to further improve the quality of synthesized portraits: $\mathbf{t}^* = \{\mathbf{t}, \mathbf{t_d}, \mathbf{t_n}\}$, where $\mathbf{t_d}$ and $\mathbf{t_n}$ mean decorative and negative prompts, respectively. Consequently, our stylized portrait generation can be rewritten as:

$$I_s = T(\phi, G(\theta, \mathbf{w}, \mathbf{v}), \mathbf{n}^*, \mathbf{t}^*), \tag{2}$$

where $I_s$ is one stylized portrait. We construct a few-shot stylized portrait dataset $\mathcal{D}_s$ using different $\mathbf{v}$ from $(P, Y)$, so $\mathcal{D}_s$ contains $i^2$ stylized portraits. The construction pipeline is summarized in Algorithm 1.

## 3.2 IMAGE-TO-TRIPLANE NETWORK

Considering the ability in generating high quality and 3D consistency image, we also utilize EG3D to synthesize 3D portrait. With the few-shot dataset, an intuitive solution is using it to fine-tune a pre-trained EG3D. However, we find that training the entire EG3D is not efficient, and the rich facial prior hidden in the pre-trained EG3D can also be damaged. In order to fully enjoy the priors, one may invert $\mathcal{D}_s$ to the $\mathcal{W}$ latent space. Unfortunately, it will limit the portrait style within the scope of training set that EG3D is pre-trained, since the triplane feature generator is not changed. As a result, we introduce I2T module to learn the mapping from $\mathcal{D}_s$ to triplane feature space. We replace the original triplane generator of EG3D with our I2T module and keep other parameters unchanged as a EG3D renderer, reaching a trade-off between training efficiency and prior utilization.

Similar to the triplane generation network of the original EG3D, I2T consists of multiple StyleGAN modulation layers (*SMLs*) and a style encoder $E$, as shown at the bottom left corner of Fig. 2. Due to the small size of $\mathcal{D}_s$, we expand the latent code $\mathbf{w}_s$ along spatial dimension (from $c$ to $c \times k \times k$) to enrich it with style and structural information. Then we adopt $E$ to learn $\mathbf{w}_s$ first, and feed it into *SMLs* to generate stylized triplane. For a particular feature layer $F^i \in \mathbb{R}^{C_i \times H_i \times W_i}$ in *SMLs*, we have the following formula:

$$F^i_{c,h,w} = \gamma^i_{h,w}(\mathbf{w}_s) \times \frac{F^i_{c,h,w} - \mu^i_{h,w}}{\sigma^i_{h,w}} + \beta^i_{h,w}(\mathbf{w}_s), \tag{3}$$

where $\mu^i_{h,w}$ and $\sigma^i_{h,w}$ represent the calculated mean and standard deviation across channel dimension, respectively. $\gamma^i_{h,w}$ and $\beta^i_{h,w}$ are learnable weight networks. It is noted that, to ensure stable feature learning, the input portrait of $E$ is always of front view, which contains richest style information compared with other viewpoint.

---

**Algorithm 1** Few-shot dataset construction

1: **Input:** $\mathbf{w} \sim \mathcal{W}$, $\mathbf{n} \sim \mathcal{N}(0, 1)$, $\mathbf{t}$, $\mathcal{D}_s = \emptyset$
2: **for** $\mathbf{v}$ in $(P, Y)$ **do**
3:   $I = G(\theta, \mathbf{w}, \mathbf{v})$
4:   $\mathbf{n}^* = Add(\mathcal{E}(I), \mathbf{n}, \tau)$
5:   $I_s = T(\phi, I, \mathbf{n}^*, \mathbf{t}^*)$
6:   $\mathcal{D}_s = \mathcal{D}_s \cup I_s$
7: **end for**

**Algorithm 2** The I2T network training

1: **Input:** $\mathcal{D}_s$, Pre-trained I2T
2: Init I2T net using the pre-trained I2T
3: **repeat**
4:   Select $I_v$, $I_f$ from $\mathcal{D}_s$
5:   Fine-tuning I2T net in $\mathcal{D}_s$ using loss:
6:     $\mathcal{L}_{\text{total}} = \lambda_{\text{rec}} \mathcal{L}_{\text{rec}} + \lambda_{\text{dr}} \mathcal{L}_{\text{dr}}$
7: **until** end of iterations

---

Although I2T network can be trained on $\mathcal{D}_s$, it still suffers from two challenges: First, $\mathcal{D}_s$ is a few-shot dataset, it has no more than 100 images for a style in practice. Second, the portraits in $\mathcal{D}_s$ have more or less differences in style, resulting in 3D inconsistency. We alleviate this problem by pre-training the I2T network on the portraits generated by EG3D prior. In particular, in each iteration of the pre-training, we randomly generate a portrait from EG3D, and record its triplane representation as $\mathbf{p} \in \mathbb{R}^{256 \times 256 \times 96}$, its front view as $I_f \in \mathbb{R}^{512 \times 512 \times 3}$. Then the I2T network is pre-trained using the following loss function:

$$\mathcal{L}_{\text{I2T}} = \mathbb{E}_{I_f, \mathbf{p}}[\|H(I_f) - \mathbf{p}\|_1], \tag{4}$$

where $H$ represents the I2T network. After some iterations, we can learn the mapping between the input portrait and its triplane representation.

**Training.** After being pre-trained, the I2T network can be quickly fine-tuned on the stylized multi-view portrait dataset $\mathcal{D}_s$, as shown in the lower part of Fig. 2. For an arbitrary view portrait $I_v$ and front view portrait $I_f$ in $\mathcal{D}_s$, we use the following image reconstruction loss:

$$\mathcal{L}_{\text{rec}} = \mathbb{E}_{I_v, I_f \in \mathcal{D}_s}[\|G^*(I_f, \mathbf{v}) - I_v\|_1 + lpips(G^*(I_f, \mathbf{v}), I_v)], \tag{5}$$

where $\|\cdot\|_1$ is the $L_1$ reconstruction loss, $G^*$ represents the new EG3D model with I2T network. $lpips(\cdot, \cdot)$ is the Learned Perceptual Image Patch Similarity (Zhang et al., 2018) loss, which calculates the distance of the latent features extracted from the VGG network. In addition to the reconstruction loss, we add the density regularization, which encourages smoothness of the density field rendered by triplane and prevents sharp or hollow portrait shapes during fine-tuning. The density regularization loss is shown as follows,

$$\mathcal{L}_{\text{dr}} = \mathbb{E}_{\mathbf{x}, \delta}[\|\sigma(\mathbf{x}) - \sigma(\mathbf{x} + \delta \cdot \mathbf{x})\|_1], \tag{6}$$

where $\mathbf{x}$ are the random sampling points in the volume rendering, $\delta$ is a small Gaussian noise, and $\sigma(\mathbf{x})$ denotes the density rendering process. Thus the final loss function is:

$$\mathcal{L}_{\text{total}} = \lambda_{\text{rec}} \mathcal{L}_{\text{rec}} + \lambda_{\text{dr}} \mathcal{L}_{\text{dr}}, \tag{7}$$

where $\lambda_{\text{rec}}$ and $\lambda_{\text{dr}}$ are loss weights.

Thanks to the pre-trained I2T, we can quickly extend the domain of I2T to stylized portraits $\mathcal{D}_s$ with only a few training iterations. The total training process is listed in Algorithm 2.

## 4 EXPERIMENTS

### 4.1 IMPLEMENTATION DETAIL

Our method is implemented in PyTorch using an NVIDIA A100. We use Adam optimizer with learning rate of 0.002 and $\beta_1 = 0, \beta_2 = 0.99$. The number of samples $i$ in few-shot dataset construction is 10. Other parameters, such as camera focal length, use the EG3D default settings. In Ip2p inference, we set the number of time step $\mathbf{s}$ in DDIM (Song et al., 2021) to 20. The degree of noise addition $\tau$ is 0.9, and the image guide and text guide weight parameters of Ip2p are set to 1.5 and 20.0. The decorative prompt $\mathbf{t_d}$ is "realistic, detail, 8k, photorealistic", then the positive prompt input for Ip2p is "turn the head into $\mathbf{t}$, $\mathbf{t_d}$". The negative prompt $\mathbf{t_n}$ is "unclear facial features, non-face objects, ugly, bad". Note that we fix $\mathbf{t_d}$ and $\mathbf{t_n}$, and only change $\mathbf{t}$ in all following experiments. Fine-tuning $\mathbf{t_d}$ or $\mathbf{t_n}$ will polish the generated results, but it's not the scope of this paper. For each text prompt $\mathbf{t}$ we randomly sample a $\mathbf{w}$ and construct the few-shot dataset according to Algorithm

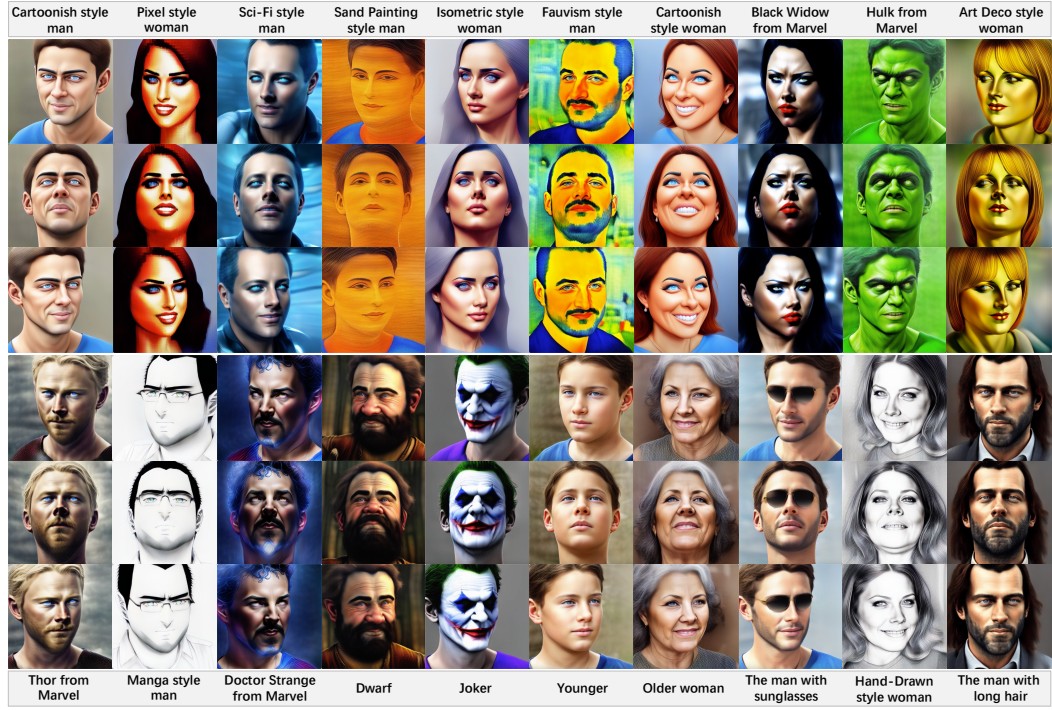

Figure 3: Multi-style and Multi-identity 3D portrait synthesis results.

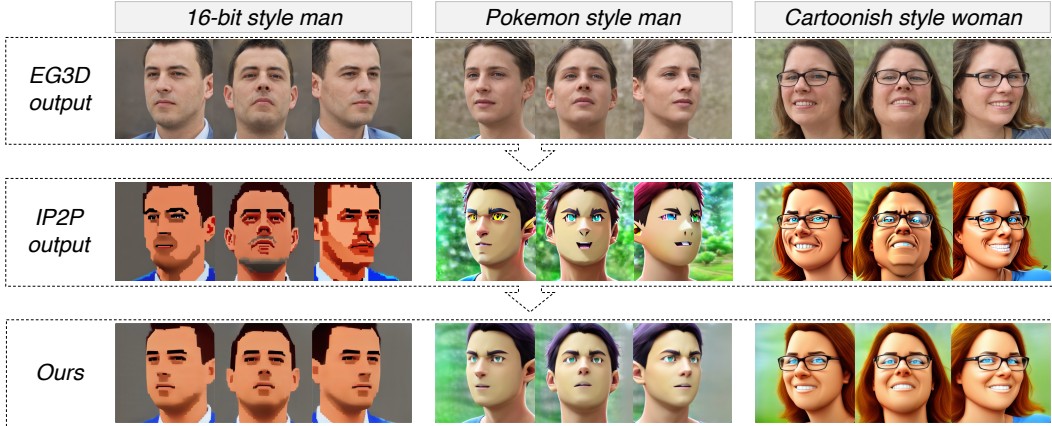

Figure 4: The outputs of different stages in our method.

1. In I2T network, The spatial dimension $k$ of $\mathbf{w}_s$ is 32. *SMLs* consists of 7 modulation layers. The I2T network is pre-trained on EG3D randomly sampled data for 100k iterations. When fine-tuning the I2T network on the few-shot dataset, the loss weights are set as $\lambda_{\text{rec}} = 10.0$ and $\lambda_{\text{dr}} = 0.2$.

## 4.2 FREE-STYLE 3D PORTRAIT SYNTHESIS

In this section, we show the 3D portrait synthesis results of our approach. As shown in Fig. 3, our method can generate diverse style 3D portrait, and the synthesized portraits are high quality and 3D consistent, proving our ability of free-style generation. Then, we display the outputs of different stage (i.e., EG3D, IP2P and final output) in Fig. 4, it can be seen EG3D outputs the original 3D portrait, IP2P will add the style into the portrait images (the facial details and consistency can not be guaranteed), our method will generate the final high quality and 3D-consistency results. In addition, we generate the 3D geometry results of different style in Fig. 5, which can accurately reflect the

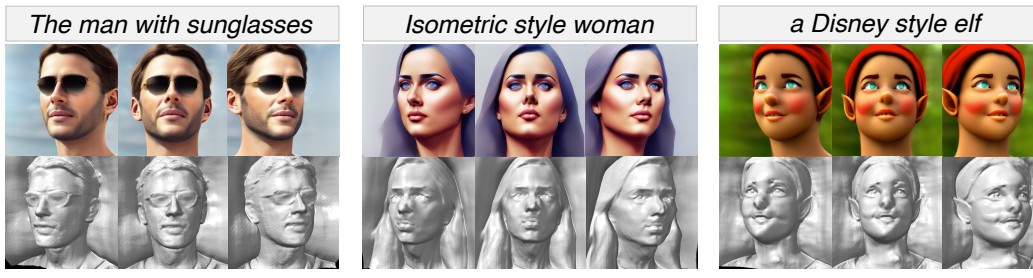

Figure 5: Visualization of 3D geometry from different stylized generation results.

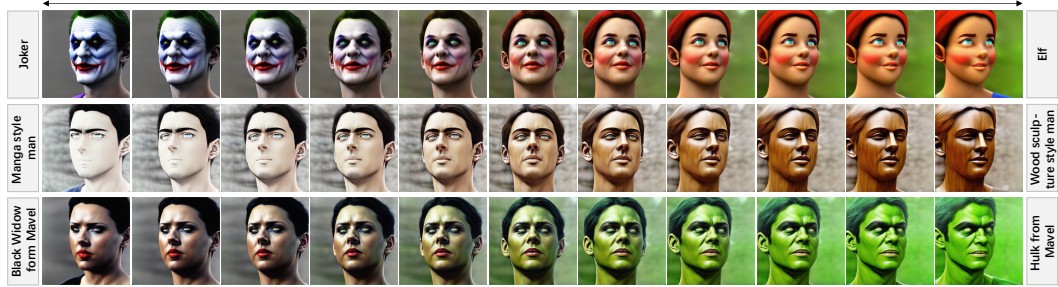

Figure 6: The 3D portrait results synthesized by mixing different styles in the $\mathbf{w}_s$ latent space.

given style. Last but not least, Fig. 6 shows the smooth style mixing results of our method. We interpolate the encoded features of two different style in the $\mathbf{w}_s$ latent space, and use the new triplane representation to synthesize 3D portraits. The results reveal the controllability and potential style editing ability of our model.

### 4.3 COMPARISON EXPERIMENTS

**Baselines.** We divide the baselines into two categories. 1) Text-to-3D. DreamFusion[1] (Poole et al., 2022) is a representative method that generates 3D images based on the text prompts. 2) Image style transfer + 3D GAN Inversion. 3DGAN-Inv (Ko et al., 2023) and HFGI3D (Xie et al., 2023) are two SOTA methods of 3D GAN inversion. We use them with Instruct-pix2pix as the baselines.

The results of our method compared with baselines are shown in Fig. 7. DreamFusion, the representative of the Text-to-3D method, is able to optimize the model to synthesize 3D portraits based on text prompts, such as "A Disney style Elf", but the generated results have low quality, while for more specific portrait styles, such as "sand painting style", no results can be synthesized. What's more, the portraits with large stylistic variations cannot be inverted well using the 3DGAN-Inv, because the 3D portraits synthesized by this method cannot escape from the domain of the pre-trained 3D-aware GAN model. The results of HFGI3D are better, but the 3D shape is destroyed after inversion, as shown in the Elf's face. At the same time, for some more difficult samples, this method cannot produce effective results, such as the last two lines. In contrast, our method is able to synthesize high quality 3D portraits that satisfy both 3D consistency and stylization.

For quantitative evaluation, we conduct a user study. We invite 30 volunteers to evaluate each method from three perspectives, namely the text-image similarity, the quality of the generated images, and the 3D consistency of the results. Each item is scored on a 5-point scale, and the average is calculated as the final result. As shown in Tab. 1, our approach achieves the best scores under each dimension. We also calculate the CLIP Score for each method shown in Tab. 1, which uses the CLIP (Radford et al., 2021) model to extract features and calculate cosine similarity between input text and generated image. Our method also achieves optimal result. In addition, our method also has a significant advantage in running time, as shown in Tab. 1. Our method is able to fine-tune model on the given text prompt in about 3 minutes, while other approaches require more time. Although

---

[1]https://github.com/ashawkey/stable-dreamfusion

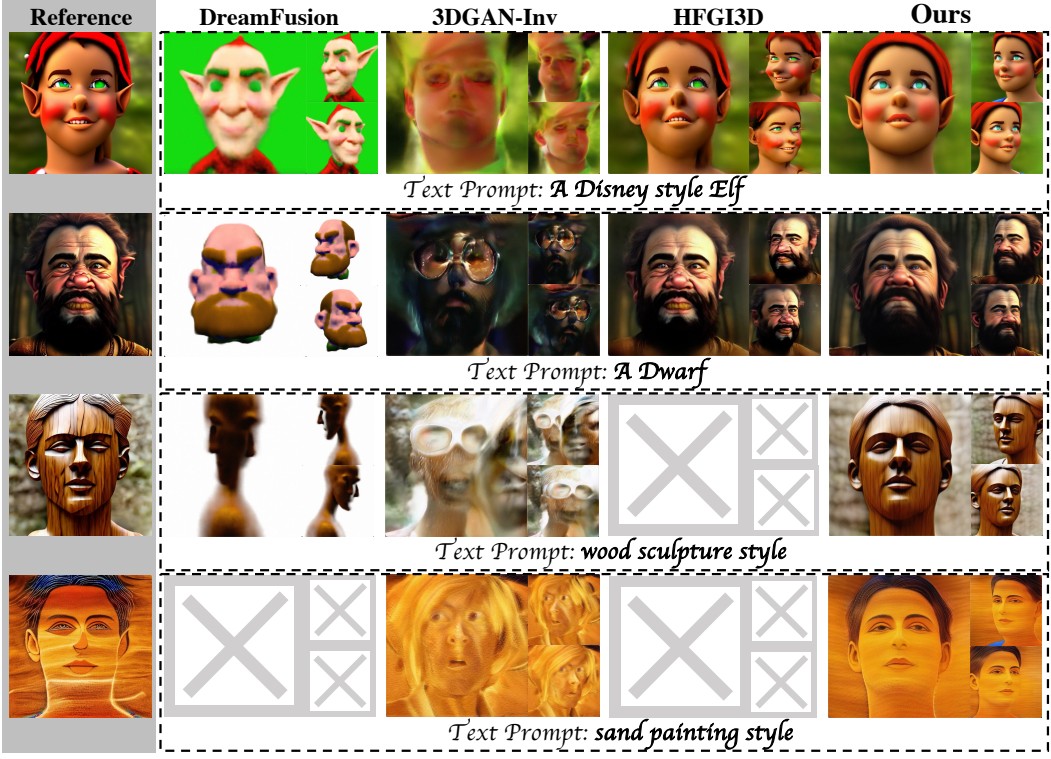

Figure 7: Qualitative comparison results between our method and baselines. Reference image represents the result after Ip2p style transformation, which is used as the input of the 3D GAN Inversion methods.

Table 1: User Study, quantitative evaluation and running time of different methods.

| method | Text-image Similarity | Image Quality | 3D Consistency | CLIP Score | Average Time |
|--------|----------------------|---------------|----------------|------------|--------------|
| DreamFusion | 2.82 | 2.24 | 3.18 | 0.304 | $\sim$ 40 mins |
| 3DGAN-Inv | 2.04 | 1.90 | 2.59 | 0.229 | $\sim$ 3 mins |
| HFGI3D | 3.40 | 3.54 | 3.67 | 0.310 | $\sim$ 10 mins |
| Ours | **4.05** | **4.27** | **4.34** | **0.332** | $\sim$ 3 mins |

3DGAN-Inv costs comparable running time to ours, its portrait generation quality is poor and fails to generate free-style portraits.

### 4.4 ABLATION STUDY

We conduct the ablation studies of our method from three aspects, which are shown in Fig. 8. First, when we do not optimize the Ip2p inference, some character prompts generate poor results, such as the example in (a) of Fig. 8. Second, we perform the ablation of the I2T network in (b) of Fig. 8. 1) When we remove the I2T network and instead add two modulation layers directly to the original EG3D triplane generator (only the modulation layers are trainable, original EG3D triplane generator is fixed), we can achieve some stylization effect. However, the generation quality is limited by the lightweight modulation layer. 2) When using our I2T network without pre-training, training I2T from scratch on the

Table 2: Quantitative results on I2T network.

| method | CLIP Score |
|--------|-----------|
| w/o I2T | 0.251 |
| w/o I2T pre-training | 0.113 |
| w/o $\mathbf{w}_s$ spatial dimension | 0.282 |
| Ours | **0.332** |

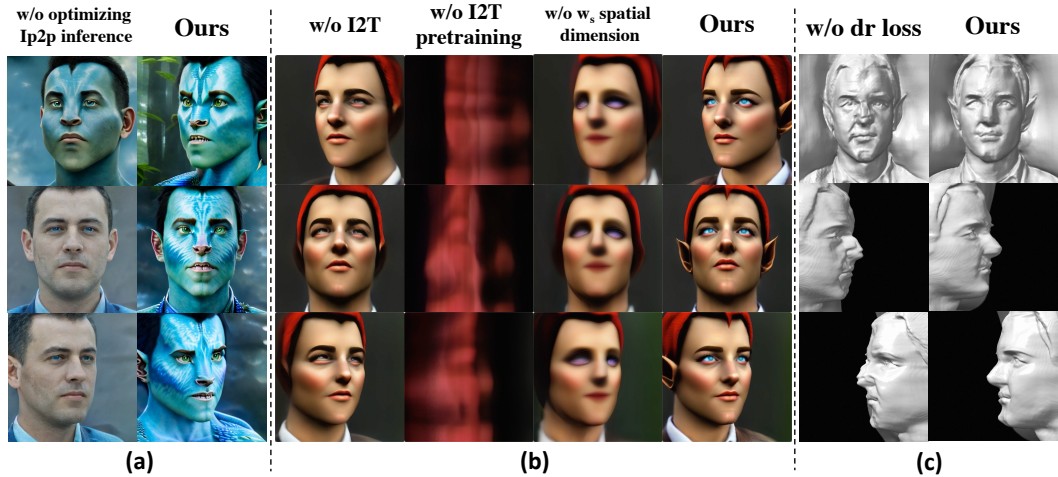

Figure 8: The ablation studies of our approach. (a) Ip2p generated results w/ or w/o inference optimization. (b) 3D portrait synthesis results in different I2T network settings. (c) 3D shape outputs w/ or w/o dr loss.

few-shot dataset will cause mode collapse. Because it is difficult to establish the mapping from the image to the triplane representation only using 3D inconsistent data. 3) When the $\mathbf{w}_s$ code of our I2T network is a vector without spatial dimension, the learned 3D portrait is blurred and lacks details. Furthermore, we calculate the CLIP Score of different ablation models in Tab. 2, and our method achieves the best result. In addition, we ablate the density regularization (dr) loss used in model training, shown in (c) of Fig. 8. When the density regularization is not used, the synthesized portrait shape will have rough surfaces.

## 5 LIMITATION

Our approach is based on two powerful pre-trained generative priors, which are the basis of our method's ability to synthesize high-quality stylized 3D portraits. At the same time, our method is limited by both priors, especially Ip2p, which is unable to achieve perfect 3D-consistent portrait stylization in different viewpoints, so the final synthesized 3D portrait of our method differs slightly from the style generated by Ip2p. Meanwhile, some stylistic changes that differ significantly from the human portrait shape, such as "Iron Man" and "Stormtrooper" in Fig. 9, result in poorer quality 3D portraits compared to human portraits. Because when Ip2p generates this type of style, the stylization effect will be more different under different views.

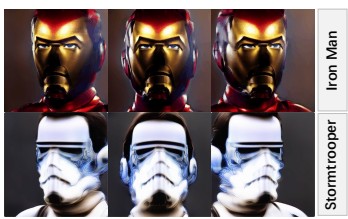

Figure 9: Some bad cases.

## 6 CONCLUSIONS

This paper proposes a novel free-style and fast 3D portrait synthesis framework. Our method is based on two pre-trained generative priors, EG3D and Ip2p. We optimize Ip2p inference in order to stylize portraits at different views more freely and stable. We can quickly construct a few-shot training set of stylized portraits using the two generative priors, and fast fine-tune the EG3D prior. Moreover, We replace the original triplane generator in EG3D with a trainable I2T network to help fine-tune EG3D more efficiently. A large number of high-quality 3D portrait synthesis results and comparison experiments with baselines show the superiority of our method.

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
