# OpenReview forum: "Free-style and Fast 3D Portrait Synthesis"
_ICLR.cc/2024/Conference — ICLR 2024 Conference Withdrawn Submission_

### Official Review · Reviewer_jiQH · 2023-10-28

**Soundness:** 2 fair
**Presentation:** 3 good
**Contribution:** 2 fair
**Rating:** 3
**Confidence:** 4

**Summary:**

This paper proposed a free-style 3D portarit synthesis method by leveraging two existing generative priors. The authors at first use modified 3D aware ED3G generator to synthesize portrait with a certain pose and then use text-guided instruct pixel2pixel for portrait stylization. Experiments shows the proposed method is able to generate stylized portraits with specific style and pose in short time.

**Strengths:**

The proposed method is flexible that can generate portraits with differet pose and styles could be easily controled by text prompts; It synthesizes results with high quality and out-performs previous methods

**Weaknesses:**

It seems to me that the proposed method is a naive combination of two existing methods (with a little modification), Thus the technical novelty and contribution are limited.
For the synthesis time, the average running time of the proposed method is 3 minutes and it's still quite long for image synthesise tasks. It may not be proper to call it "fast".
For the user study part, it seems the authors didn't mention how many samples are shown to each participant.
In the comparison, the authors didn't show the results of simple applying the EG3D and then Instruct pix2pix

**Questions:**

Why are EG3D and Instruct pixel2pixel used in the proposed method? Is there any insight or analysis behind or just because they are the state-of-the-art 3D face image synthesis and image stylization method?
How would the authors pursuade that the proposed method is noval and the technical contribution is solid, instead of a combination of existing works than achieves good performance on a certain application?
What's the connection between these two methods? is there any interaction between them or just use modified version for separate inference?

---

### Official Review · Reviewer_8A9M · 2023-10-28

**Soundness:** 2 fair
**Presentation:** 2 fair
**Contribution:** 2 fair
**Rating:** 5
**Confidence:** 3

**Summary:**

The paper proposes a text based stylized 3D portrait synthesis method. The text prompt defines the style. The proposed method generates a dataset using EG3D and instruct pix2pix. Both quantitative and qualitative results are good.

**Strengths:**

- The experimental results are good, compared with baseline approaches. The video results are also good.

**Weaknesses:**

- The paper mentions that existing 3D GAN inversion methods "cannot jump out of the pre-trained 3DGAN prior and
cannot synthesize out-of-distribution portraits". However, it is not the problem of the method. It is the problem of the data. If the baseline approaches are trained with stylized datasets, will they outperform the proposed method?

- The novelty seems limited. Data generation is a straight forward combination of EG3D and instruct pix2pix. The inversion module is a common 3D GAN inversion architecture.

**Questions:**

I would like to see the authors address my concerns mentioned in the weakness section.

---

### Official Review · Reviewer_1GQB · 2023-10-30

**Soundness:** 3 good
**Presentation:** 4 excellent
**Contribution:** 2 fair
**Rating:** 5
**Confidence:** 4

**Summary:**

This paper proposes a method for free-style and fast 3D portrait synthesis. The key contributions are:

+ The authors combine two generative priors - EG3D (a 3D-aware GAN) and InstructPix2Pix (a text-guided image editor) - to quickly construct a few-shot training set with a desired style specified by a text prompt.
+ The authors optimize the inference process of InstructPix2Pix to generate more stable/consistent stylized portraits from different viewpoints (quickly construct a few-shot training set).
+ The authors replace the triplane generator of EG3D with a trainable Image-to-Triplane (I2T) module and only fine-tune I2T on the few-shot dataset to achieve fast stylized 3D portrait synthesis while retaining useful priors from EG3D.

In summary, the key innovation is efficiently combining two powerful generative priors to enable free-style 3D portrait synthesis with both quality and speed. The optimized I2T module balances retaining priors and fast fine-tuning.

**Strengths:**

This paper presents a fast 3D portrait synthesis framework by rationally combining and optimizing two state-of-the-art generative models. The framework can use textual hints to specify specific styles. Following are its main strengths:

+ Combining EG3D with InstructPix2Pix to optimize inference to quickly build a stylized dataset of a small number of faces.

+ Replacing the EG3D triplane generator with a fast-trainable Image-to-Triplane (I2T) module for efficient fine-tuning.

+ The paper is clearly written and easy to follow. The methodology explains each component well, with useful diagrams. The experiments systematically validate the approach.

+ The synthesized 3D style portraits have a good quality from the given metrics. The structural consistency of the views is maintained, while the style corresponding to the text is obtained.

**Weaknesses:**

Here are some potential weaknesses of this paper on free-style and fast 3D portrait synthesis:

The overall contribution is not strong. The method of combining EG3D and InstructPix2Pix to build a style data set has been used by previous papers, such as DATID-3D (https://arxiv.org/abs/2211.16374). And here are compared and introduced in related work and experiments. There are also some additional text-guided style methods based on EG3D, such as Diffusion Guided (https://arxiv.org/abs/2212.04473). There is no need to analyze methods and compare results in this paper.

The ablation of the Image-to-Triplane module (I2T) proposed in this article cannot fully demonstrate its superiority. I think fine-tuning the triplane generator directly can also produce nice stylistic results. At the same time, only fine-tuning the tri-plane generator only changes the tri-plane features of EG3D. I think it is similar to using the I2T module in terms of retaining the prior knowledge of the model.

Optimizing Instruct-pix2pix inference also seems to use an existing method.

Quantitative evaluation is mainly limited to CLIP similarity and user studies. More rigorous objective quantitative benchmarks needed.

Limited diversity in synthesized identities and shapes - mainly seen human portraits. Other objects/styles may be more challenging.

**Questions:**

Here are some questions and suggestions for the authors:

The InstructPix2Pix optimization helps but doesn't fully solve multi-view inconsistencies. This will have a direct impact on the quality of the generated data set and subsequent training, and whether there are other ways to supervise the consistency of different perspectives and styles.

More comparative experiments need to be added, at least more similar work should be included in the comparison, such as DATID-3D.

The identities and shapes synthesized seem largely limited to human portraits. How well does your approach generalize to other objects, geometries, and more extreme stylizations? (EG3D is not limited to natural faces)

Could you expand the ablation study? e.g. optimizing different components, or more styles.

The contribution of the paper is currently insufficient. Please provide more evidence to prove the contribution of the revised paper.

---

### Official Review · Reviewer_W4GE · 2023-10-31

**Soundness:** 2 fair
**Presentation:** 2 fair
**Contribution:** 2 fair
**Rating:** 5
**Confidence:** 4

**Summary:**

This paper introduces an innovative pipeline for stylized 3D portrait synthesis. The approach leverages the power of two pre-trained generative models, namely EG3D and instruct pix2pix (ip2p). To enhance the versatility and stability of generating stylized portraits from different angles, the authors refine the ip2p inference and effectively construct a few-shot training dataset of stylized portraits. Subsequently, they expedite the fine-tuning of the EG3D prior. The results are compared against various baseline methods, demonstrating the superior performance through a range of 3D portrait synthesis examples.

**Strengths:**

The pipeline exhibits a technical foundation, yielding visually plausible results. Furthermore, the generated video maintains a high level of 3D consistency. The concept of integrating instruct pix2pix into a 3D generative framework for achieving stylization is particularly intriguing.

**Weaknesses:**

My primary concern lies in the fact that this paper predominantly relies on existing components and training losses, including pre-trained models like EG3D, an instruct pix2pix image stylization model, and an image-to-triplane encoder. The pipeline appears to directly combine these three components.

To address this, it would be beneficial to conduct a thorough comparison with a state-of-the-art method such as "Live 3D Portrait: Real-Time Radiance Fields for Single-Image Portrait View Synthesis (ACM Transactions on Graphics, SIGGRAPH 2023)." This method also demonstrates one-shot stylized 3D head generation with a fixed EG3D generation, eliminating the need for further fine-tuning.

Additionally, in Figure 7's second row, it seems that HFGI3D exhibits better reconstruction quality compared to the proposed method, with superior expression preservation. It's worth noting that there are missing results for the last two examples from HFGI3D, which raises questions about their performance in those cases.

**Questions:**

Please refer to above.